# Oscillation Flow Dam Operation Method for Algal Bloom Mitigation

**Jungwook Kim** [1] , **Jaewon Kwak** [2,\*], **Jung Min Ahn** [1], **Hongtae Kim** [1], **Jihye Jeon** [1] and **Kyunghyun Kim** [1]

1   Water Quality Assessment Research Division, Water Environment Research Department, National Institute of Environmental Research, Incheon 22689, Korea; rlawjddnr1023@gmail.com (J.K.); jahn@korea.kr (J.M.A.); htkim8@korea.kr (H.K.); jhjeon16@korea.kr (J.J.); matthias@korea.kr (K.K.)
2   Forecast and Control Division, Han River Flood Control Office, Seoul 06501, Korea
\*   Correspondence: firstsword@korea.kr

**Abstract:** Green algae play an important role in ecosystems as primary producers, but they can cause algal blooms, which are socio-environmental burdens as responding to them requires water resources from dam reservoirs. This study proposes an alternative for reducing algal blooms through dam operation without using additional water resources. A novel oscillation flow concept was suggested: oscillating discharge of dam for irregular flow. To examine its effect, the Environmental Fluid Dynamics Code—National Institute of Environment Research (EFDC-NIER) model was constructed and calibrated for the Namhan River, South Korea, from downstream of the Chungju Dam to downstream of Gangcheon Weir. The water quality in the study area were simulated and analyzed for August 2019, which is when the largest number of harmful cyanobacteria had been reported in recent years. Our results showed that the oscillation flow produced significant variance of flow velocity, and algal bloom density in the Namhan River was reduced by 20–30% through the operation of the Chungju Dam. However, further study and investigation are required before practical application.

**Keywords:** algal blooms; dam operation; EFDC-NIER





## 1. Introduction

Algae play an important role in ecosystems as primary producers. However, algal blooms in rivers result from mass propagation of algae and have negative socio-environmental impacts such as increased turbidity, odor, oxygen depletion, sediment formation, degradation of water quality, and increased levels of toxic substances [1]. Because of climate change, green algal blooms and subsequent water quality problems have become more frequent and have increasingly entered the public consciousness. Consequently, the Ministry of Environment (ME) of South Korea recently included algae in their seven-day water quality forecasting to predict changes in water quality in advance; this will help water managers preemptively respond to water quality problems [2].

Because algal blooms significantly impact water usage and environment, many studies have been conducted to examine methods for controlling algal growth. In general, these methods can be divided into physical, chemical, biological, or combined methods according to the control mechanism [3]. Physical methods typically aim to create an environment in which it is difficult for algae to grow by accelerating internal and external water circulation or performing filtration before and after the occurrence of green algae. The aeration, dilution, and circulation of water are examples of physical methods [4,5]. Chemical methods use substances to precipitate and insolubilize algae; they typically precipitate green algae through chemical reactions. A representative example is the use of coagulants during algal blooms [6,7]. Biological methods control algae using organisms that compete with algae for the nutrients required by them or organisms that feed on algae directly. The use of artificial wetlands after sewage treatment plant or freshwater shellfish circumfusion in

rivers are examples of biological methods [8–10]. Finally, combined methods combine physical, chemical, or biological methods [11].

In South Korea, the methods to control algal growth are applied according to their advantages and associated conditions. One of these conditions is the duration of effectiveness. Most green algae control technologies have exhibited short-term algae inhibition effects of less than two weeks, but algae bloom typically occurs throughout the summer and fall seasons in South Korea [11]. Therefore, treatment must be applied repeatedly [12], limited the practical effect on algae bloom in rivers, dams, and reservoirs [13,14]. Another condition associated with methods for controlling algal growth is the concentrated locations of algae bloom. The algae bloom in Korea is mostly concentrated in reservoirs created by dams or in rivers downstream of dams. Considering past studies that have shown that lakes generated by dams have an increased likelihood of containing algae [15,16], dams and their operation would likely be a key factor in algae growth and bloom [17]. Therefore, controlling algae in rivers downstream of dams has been researched as a part of dam operation [18]. For instance, Tomczyk et al. [19] reported that the operation of hydropower should have a harmful effect on water quality and algae bloom in downstream waters, and Yoshioka and Yaegashi suggested a stochastic method to determine optimal dam operation for algae control downstream of dams [20]. Li et al. claimed that the flow rate and velocity are major management points in controlling algae bloom in rivers downstream of dams [21].

In South Korea, the flushing-out method has mainly been used for controlling algal blooms considering the flow rate as a major factor in algal growth and spread [22]; this method involves flushing algae in rivers downstream by discharging large amounts of water from dams within a short period [23,24]. Along with controlling green algae, flushing flow has also been effectively used for controlling water pollution caused by pollutant leakages and salinity intrusions [25–27]. However, this method uses excessive amounts of water, making it necessary to consider its socio-economic and ecological impacts [28]. In areas where water supply significantly relies on dams, such as in Korea [29], using dam storage for flushing flow is a major burden on water resources. Flushing flow has been conducted in the four major rivers of Korea to improve water quality in the downstream areas of nine multipurpose dams, but its effects could not be clearly explained because of a lack of flushed amount [30,31]. Thus, effective methods for controlling algae bloom without additional usage of water resources are essential for continuous algae control.

This study aims to present oscillation flow according to the change in flow rate and velocity as a dam operation method to mitigate algal blooms in rivers downstream of dams without using dam storage. To this end, we simulated changes in water quality in response to dam operation in the area of the Namhan River downstream of the Chungju Dam until the Gangcheon Weir. We constructed an Environmental Fluid Dynamics Code—National Institute of Environment Research (EFDC-NIER) model for the study area, which is currently used for water quality forecasting by the South Korean Government. Using this model, we simulated the water quality in the study area for August 2019, the period for which the largest number of harmful cyanobacteria had been reported in recent years. We then divided the results into five cases for comparison and discussed the results and recommendations for further research.

## 2. Materials and Methods

### 2.1. Study Area

The study area consisted of a section of the Namhan River from the downstream area of the Chungju Dam to the Gangcheon Weir (Figure 1). The Chungju Dam is close to the cities of Chungju and Jecheon. It was completed in 1985 for the multipurpose utilization of water resources in the Han River and provides 3.380 billion m$^3$ of water and 844 million kWh of electricity to the Seoul metropolitan area each year [32]. In the approximately 57-km section from the downstream area of the Chungju Dam to the Gangcheon Weir, the Namhan River joins the Dalcheon river at Jungangtap-myeon, Chungju, and it flows to the Gangcheon Weir via the Chungju Regulation Reservoir and the Seom river Junction. Since

the Namhan River has a relatively larger pollution load than the Bukhan River and its major tributaries, such as Gyeongancheon, it has the largest impact on the water quality and green algae in the downstream areas of the Han River, causing algae problems every year [33]. To construct the EFDC model of the Chungju Dam and Namhan River, digital terrain, water quality, and hydro-meteorological data were essential. A 1:1000 digital topographic map provided by the National Geographic Information Institute [34] provided terrain data used to construct a digital elevation model (DEM) with a resolution of $10 \times 10$ m. The DEM was then corrected using the channel geometry of the River Master Plan and the riverbed monitoring results. Water quality data were collected from the Water Environment Information System [35]. The hydro-meteorological and dam operation data required for the model were collected from the Water Resource Management Information System [36] and the information page of K-water [32]. All hydrological data were collected from the official annual hydrological report for each year [36], and the abnormal data were removed using correction methods such as flow duration and rainfall mass curves.

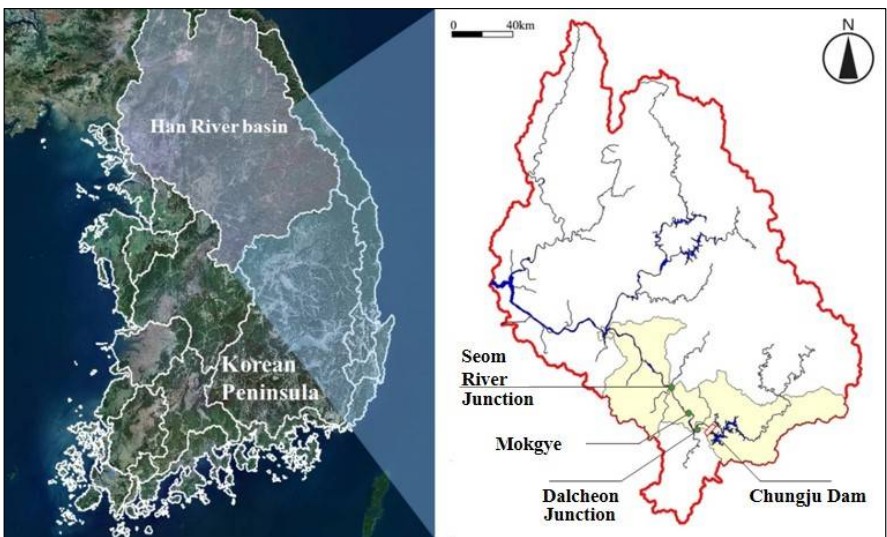

**Figure 1.** Study Area.

*2.2. EFDC-NIER Model*

Application of Equations for General Water Quality

The EFDC model is a three-dimensional water quality model that has been used worldwide to analyze hydraulic and water quality behavior in rivers, lakes, estuaries, and seas. The water quality variables of EFDC mainly consist of carbon, nitrogen, phosphorus, and silicon cycles with a focus on phytoplankton, dissolved oxygen, and COD. Phytoplankton is divided into three colonies (diatoms, green algae, and cyanobacteria) that consider seasonal transitions. However, since algae groups with different occurrence and behavioral characteristics coexist in the same colony, the conventional EFDC model has limitations in reproducing the rapid occurrence and complex species transition of algae [37]. *Microcystis*, a cyanobacteria, causes harmful algal blooms in summer in Korea. The EFDC model has been improved to allow the simulation of the transition of a specific species and has been officially used as a water quality prediction model in major river sections in Korea [38].

Since 2010, the National Institute of Environmental Research (NIER) of the Ministry of Environment (ME) improved the EFDC (20100328 version) source code to make it more suitable for South Korea's environment. The improved model was named EFDC-NIER. It is equipped with novel functions, such as the weir function for major domestic rivers, consideration of multiple algal species, and functions for simulating changes in bottom-layer nutrient flux; these functions also consider the vertical migration mechanism of cyanobacteria, cyst generation and germination mechanism, wind stress, and oxidation/reduction conditions (Figure 2). The EFDC-NIER model also allows for analysis of the flow rate and

water level in response to artificial hydraulic structures, such as the multifunctional weirs and dams developed as part of the Four Major Rivers Restoration Project [39].

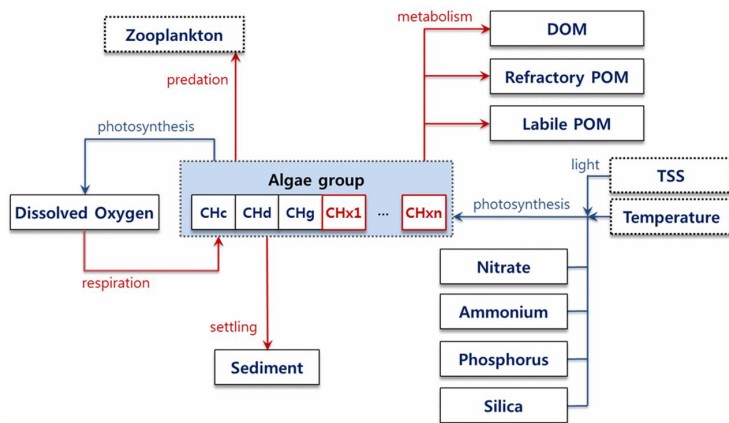

**Figure 2.** Schematic of the reactions among multiple algal species in EFDC-NIER (CHc: Cyanobacteria, CHd: Diatoms, CHg: Green algae, CH1~CHx: Multiple algal species) [39].

To perform water quality modeling using EFDC-NIER, water quality observation data for total nitrogen (TN) (mg/L), nitrate nitrogen ($NO_2-N$) (mg/L), ammonia nitrogen ($NH_2-N$) (mg/L), total phosphorus (TP) (mg/L), water temperature (°C), dissolved total nitrogen (DTN) (mg/L), dissolved total phosphorus (DTP) (mg/L), phosphate ($PO_4-P$) (mg/L), chlorophyll-a (mg/m$^3$), biochemical oxygen demand (BOD) (mg/L), chemical oxygen demand (COD) (mg/L), dissolved oxygen (DO) (mg/L), and total organic carbon (TOC) (mg/L) were required. Using these data, refractory particulate organic carbon (RPOC), labile particulate organic carbon (LPOC), dissolved organic carbon (DOC), refractory particulate organic phosphorus (RPOP), labile particulate organic phosphorus (LPOP), dissolved organic phosphorus (DOP), refractory particulate organic nitrogen (RPON), labile particulate organic nitrogen (LPON), organic matter decomposition rate (1/day) in the BOD bottle (Kdbot), and dissolved organic nitrogen (DON) were estimated by applying the fractions suggested by NIER [40]. As shown in Table 1, the concentrations in the time-series data, which were used as the input data of the model, were estimated by applying conversion equations to the observed water quality data [37]. Detailed descriptions of the parameters of the EFDC-NIER model are provided in the Supplementary Materials.

**Table 1.** Conversion equations for water quality observation data to be used as input data in the EFDC-NIER model.

| Series | Water Quality Variable | Modeling Variable | Input Data Equations |
|---|---|---|---|
| Carbon | TOC | RPOC | $= (DOC - OC) \times 0.5$ |
| | | LPOC | $= (DOC - OC) \times 0.5$ |
| | | DOC | $= (BOD - AOD_5 - NOD_5)/(1 - e^{-5 \times Kdbot}) \times (12/32)$ |
| Nitrogen | TN $NH_4-N$ $NO_3-N$ DTN | RPON | $= (TN - Algae\ Nitrogen - DTN) \times 0.5$ |
| | | LPON | $= (TN - Algae\ Nitrogen - DTN) \times 0.5$ |
| | | DON | $= DTN - NH_4 \_ NO_3$ |
| | | $NH_4$ | $= NH_4$ |
| | | $NO_3$ | $= NO_3$ |
| Phosphorous | TP $PO_4-P$ DTP | RPOP | $= (TP - Algae\ Phosphorus - DTP) \times 0.5$ |
| | | LPOP | $= (TP - Algae\ Phosphorus - DTP) \times 0.5$ |
| | | DOP | $= DTP - PO_4$ |
| | | $PO_4$ | $= PO_4$ |

*2.3. Dam Operation in Consideration of Algae*

Dam operation for algal control has focused on flushing flow because studies have reported that the flow in rivers has a significant impact on algal growth or spread [22,23]. However, the social, economic, ecological, and environmental impacts of flushing flow should be considered [28]. In addition, the effectiveness of flushing cannot be guaranteed if a considerable amount of stored water is not discharged and the duration of the effect is relatively short [30]. Therefore, conventional flushing flow is not an efficient dam operation method for algal control. Algae growth in rivers are mainly influenced by nutrients, temperature, light, stable conditions, and turbidity [41]. Among these factors, temperature and light are seasonal and are thus difficult for humans to control. It would in theory be possible to block the inflow of nutrients and turbidity, but these are difficult to artificially control in rivers in the presence of green algae. Therefore, river stability is the only factor that can be artificially controlled. Flushing flow is an algal control method that increases flow velocity to influence river stability and control algae. For example, Byeon et al. [11] determined that algae prefer stagnant lakes, reservoirs, and rivers with flow velocities of 0.2 m/s or less, where the residence time of the water is relatively long. Under these conditions, the nutrients required for algal growth can be transferred easily, thus leading to the overgrowth and accumulation of algae. Li et al. [42] reported that (1) chlorophyll-a concentration and flow velocity are inversely proportional in lakes, (2) turbulent flow severely inhibits the growth of phytoplankton and affects species composition, and (3) there is no threshold velocity that can be commonly used to control algal growth. Li et al. [42] also found that (1) there is no uniform threshold flow velocity for algal bloom control in rivers, (2) there are significant differences in the river hydrological/hydraulic conditions between years with and without algal blooms, and (3) the joint operation of cascade reservoirs and diversion projects is an effective method to prevent and control algal blooms in regulated lowland rivers. Therefore, there is not exist a uniform threshold flow velocity.

In this study, the following hypothesis was established considering the above findings: "algae will be reduced if turbulent or similar irregular flow occurs in a river according to periodic changes in flow rate and velocity."

Usually, dams discharge a certain amount of water considering the water usage downstream and the supply of instream water (Figure 3a), and the flow rate and flow velocity downstream of dams remain generally constant. When flushing flow is required for flood control and as a response to water pollution, a large amount of stored water is discharged by opening sluice gates (Figure 3b). As a result, the flow rate and velocity significantly increase in the downstream during flushing flow but become constant again when the amount of discharge decreases, and its effect also quickly disappears. If the turbulent or similar irregular flow will have lowered algae growth without consuming additional water resources, effective and continuous algae control downstream will also be possible. To evaluate the effects of oscillation flow according to the periodic change in flow rate and velocity on algal blooms, as hypothesized, oscillated discharge from the dam was proposed; it refers to discharging water from a dam in a regular vibration pattern by differentiating the constant discharge to cause changes in flow rate and velocity downstream. Considering that a low degree of water level increase and a significant increase in flow velocity were observed when flushing flow was performed in the existing dams and weirs [43–45], oscillation flow is expected to change the flow in the downstream area of a dam as a result of the significant change in flow velocity.

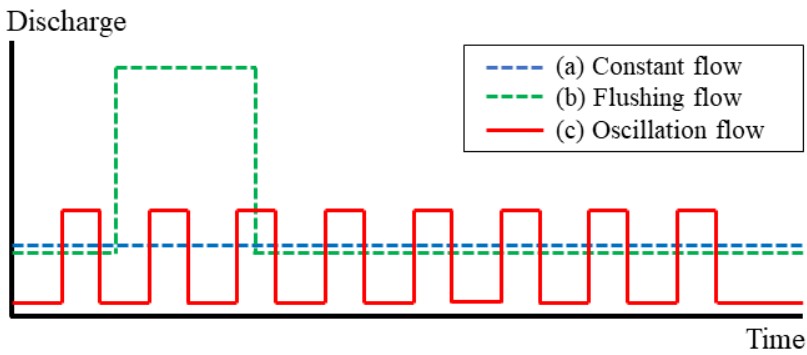

**Figure 3.** Concept of oscillation flow.

## 3. Results

### 3.1. Construction and Verification of the EFDC-NIER Model

The EFDC-NIER model of the Han River was constructed to investigate the green algae reduction effect of oscillation flow on the waters downstream of the dam. The model of the Han River consisted of the main stream section from the Chungju Dam to the Paldang Dam and some of the main streams of the Bukhan River (Cheongpyeong Dam-Paldang Dam section) (Figure 4). The model was composed of a grid with 3121 horizontal cells to reproduce the spatiotemporal diversity of water quality and algal blooms; the main boundary conditions applied are shown in Table 2. Specifically, meteorological data were obtained from the Yangpyeong weather station of the Korea Meteorological Administration (KMA), which serves the middle of the Han River. The 14 main tributaries, including the upstream boundary of the Chungju Dam, were established and considered as the boundary conditions of the inflow stream using the observed water quality data for seven-day intervals. The downstream boundary of the model was established as the Paldang Dam. Pollution sources in the main stream were established using the data of five sewage treatment plants that are connected and flow into the Han River. All the boundary conditions were matched with official forecasting models for water quality in South Korea.

**Table 2.** Main boundary condition of EFDC-NIER model in Han River.

| Main Boundary Condition | Contents |
| --- | --- |
| Weather station | Yangpyeong weather station |
| Inflow stream | Chungju Dam, Dalcheon, Yeongdeokcheon, Angseongcheon, Seom river, Cheongmicheon, Yanghwacheon, Bokhacheon, Heukcheon, Cheongpyeong Dam, Mukhyeoncheon, Gyeongancheon, Guuncheon, Byeokgyecheon |
| Downstream boundary condition | Paldang Dam |
| Pollution source (TMS) | Yeoju STP, Yangpyeong STP, Ganghwa STP, Gwangdong STP, Yangseo STP |

The ME performs model correction based on the number of cyanobacteria cells in the multifunctional weir section when implementing water quality forecasting. Similarly, water quality and algae-related parameters were calibrated for our EFDC-NIER model to examine the occurrence of green algae in the Namhan River upstream of the Gangcheon Weir. Sufficient and reliable observation data are required for calibration, and the multifunctional weir is one of the most reliable sources owing to continuous quality control for water quality forecasting and modeling [38]. Therefore, the model was calibrated based on the observed data from the section between the Seom River Junction and the upstream and downstream areas of the Gangcheon Weir. Specifically, these parameters were calibrated in the constructed model: growth rate of each algal species; the half-saturation constants of nitrogen and phosphorus; the carbon content–chlorophyll ratio; the optimum, minimum, and maximum temperatures for each algal species; the constant for the ratio of phosphorus used by the algal species; and the minimum and maximum densities for each algal species.

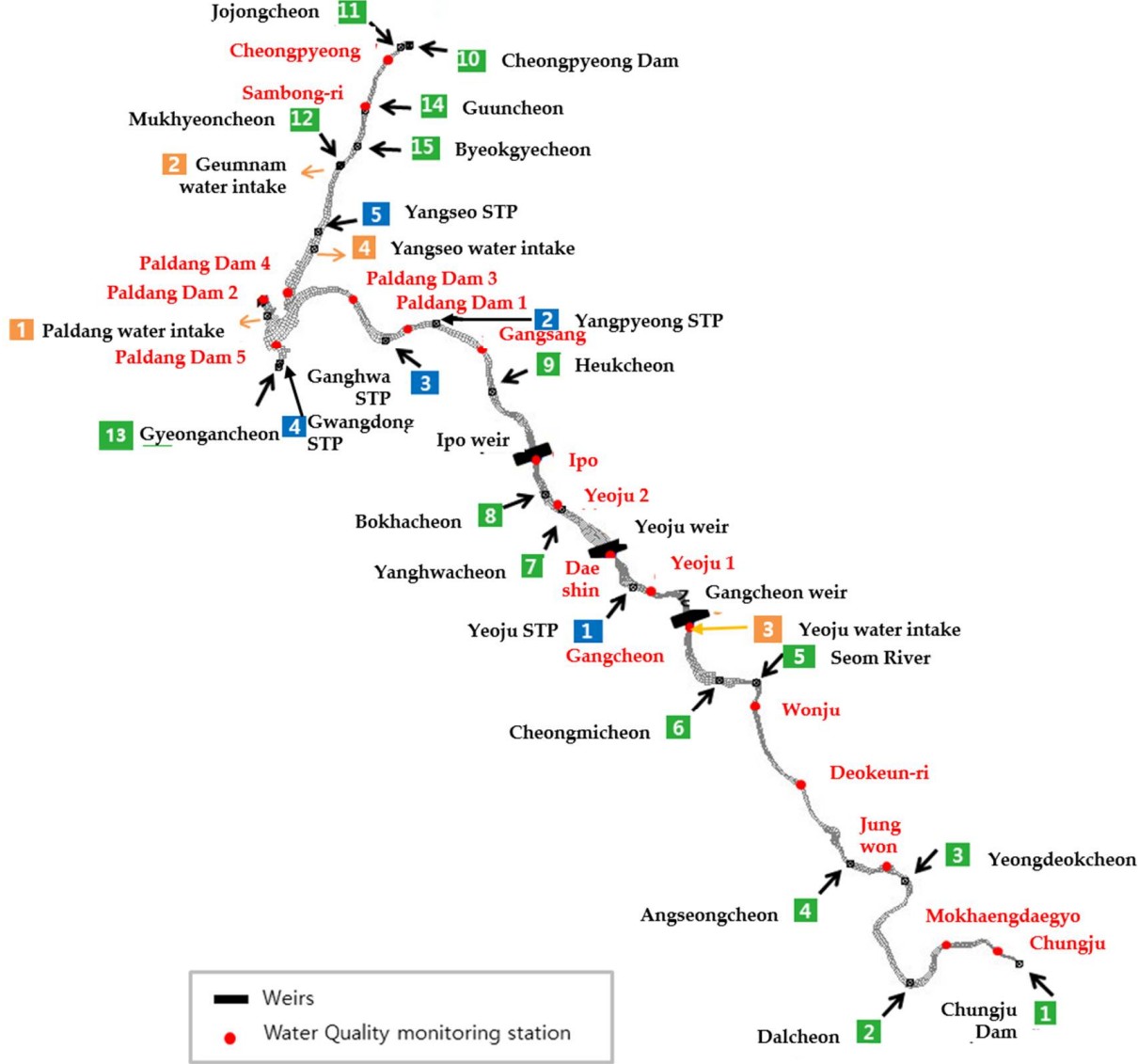

**Figure 4.** EFDC-NIER grid for the Han River(Green: Inflow stream, Orange: Water intake, Blue: Pollution source).

The calibration of the model was conducted according to the observed values for the water level and temperature, which are the main drivers of changes in the concentration of cyanobacteria. For TP and TN, which are important factors in the growth of cyanobacteria, the calibrated results were also good—they were close to the observed results (Table 3). However, in the cases of BOD and chlorophyll-a, which are the typical indicators of algal blooms, there were relatively poor results. This seems to be due to the interval of observation data, which were measured every seven days. The results of biochemical reactions such as BOD and chlorophyll-a were estimated via complex mechanisms and interactions in the model; thus, the data with 7-day intervals may not be sufficient for the complex mechanisms required by the modeling. Therefore, the calibration process was conducted as much as possible within a range that does not abnormally simulate natural phenomena (Figure 5, Table 3).

**Table 3.** Calibration result of EFDC-NIER.

|  |  | Water Level | Water Temperature | Chl-A | BOD | TP | TN |
|---|---|---|---|---|---|---|---|
| Gangcheon | $R^2$ | 0.93 | 0.99 | 0.28 | 0.41 | 0.63 | 0.78 |
|  | RMSE | 0.62 | 0.88 | 13.74 | 0.52 | 0.11 | 0.89 |
| Yeoju | $R^2$ | 0.89 | 0.99 | 0.17 | 0.30 | 0.58 | 0.79 |
|  | RMSE | 0.60 | 0.87 | 20.42 | 0.42 | 0.09 | 0.81 |

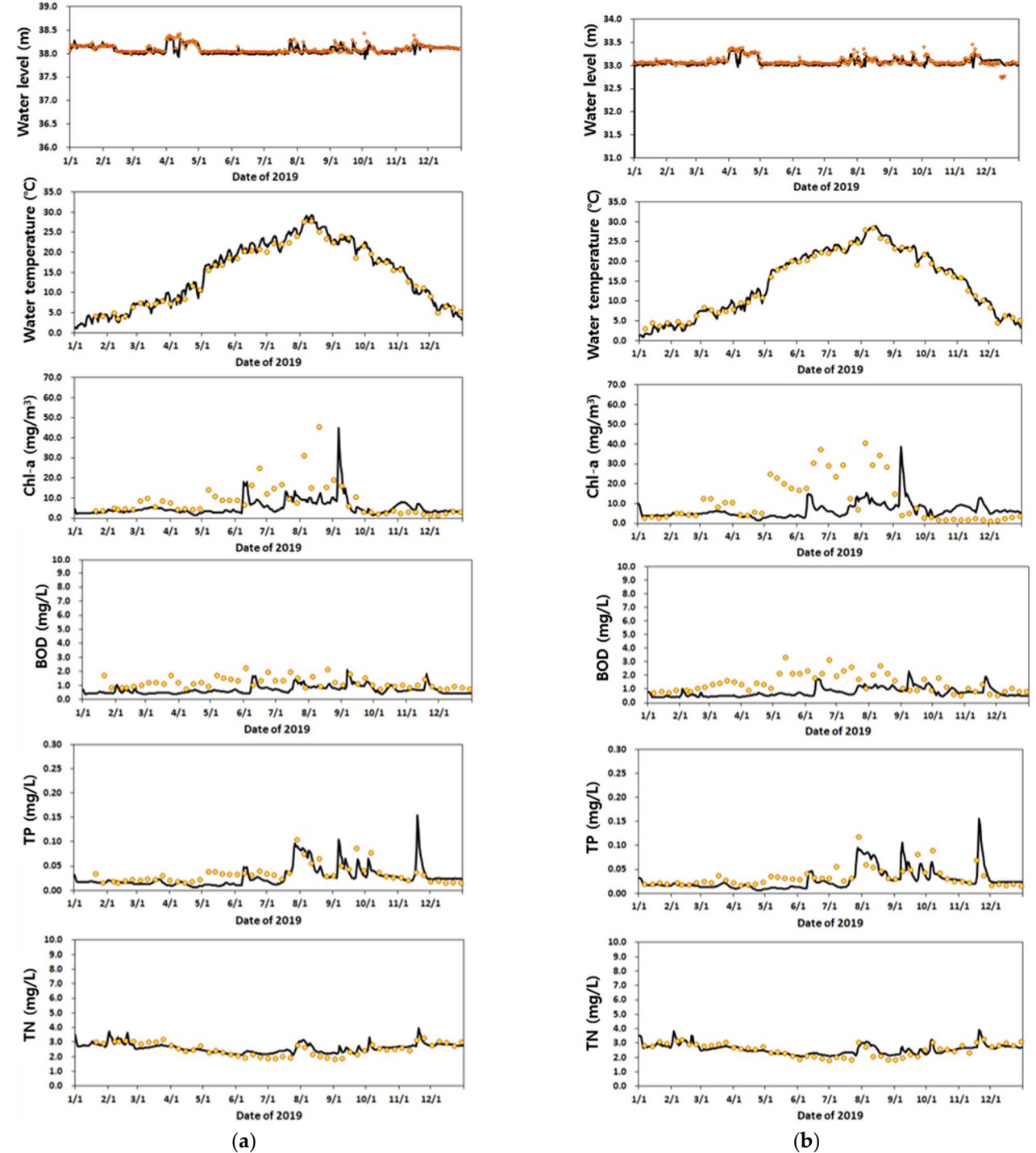

**Figure 5.** Calibration result for EFDC-NIER. (**a**) Upper reach of Gangcheon Weir. (**b**) Upper reach of Yeoju Weir.

### 3.2. Simulation of the Water Quality Change Due to Oscillation Flow

The concept of oscillation flow was applied to the EFDC-NIER model of the Namhan River, and its effect was simulated. The target period was August 2019, which was the month with the most algal blooms during the last five years, and the simulation was performed from July for model stabilization. All the data, parameters, and boundary conditions of the model are the same as the water quality forecasting by ME, except for the dam operation (oscillated discharge) of the Chungju Dam. For this parameter, the daily discharge amount of the dam observed in August 2019 was uniformly distributed every hour in the Obs case (uniform discharge), and over four, six, and eight hours for the oscillated discharge cases 1–3. The discharge range changed from 38 to 63 m$^3$/s for the Obs case, and from 0.0 to 376.0 m$^3$/s depending on each oscillated case hourly. In addition, the flushing flow used for algal blooms mitigation was also considered and compared to other cases. The case of flushing flow was based on 1 August 2019, when the number of cyanobacteria cells was large, and it was assumed that approximately 30.20 million m$^3$/s was additionally discharged for seven days. Table 4 shows the five cases considered in this study.

**Table 4.** Discharge conditions of oscillation and flushing flow.

| Item | Hourly Discharge Rate (m$^3$/s) | | | | |
|---|---|---|---|---|---|
| | Obs | Case #1 | Case #2 | Case #3 | Flushing Case |
| Discharge amount | 38.0~62.7 | 0.0~376.4 | 0.0~250.9 | 0.0~188.2 | Obs +50 m$^3$/s |
| Discharge time (h) | 0~24 (24 h) | 12~16 (4 h) | 11~17 (6 h) | 10~18 (8 h) | 168 h (from 1 August) |
| Total amount (Million m$^3$) | 131.4 | 131.4 | 131.4 | 131.4 | 161.7 |

Under the above Chungju Dam discharge conditions, the study area and section were simulated for the changes in cyanobacteria and flow velocity, and the results were reviewed at the three main points in the Namhan River (Mokgye, Seom River Junction, and Gangcheon Weir) (Figures 6 and 7 and Table 5). The results showed that flow velocities of 55, 105, and 4.5 cm/s were observed at Mokgye, Seom River Junction, and Gangcheon Weir, respectively, during conventional dam operation (uniform discharge). The standard deviation of the flow velocity ranged from 3 to 8 cm/s, with a deviation of approximately 5% compared to the average flow velocity, and the water level also deviated by 3–4 cm from the average water level. This indicates that the conventional operation of Chungju Dam generates a constant flow that does not cause significant changes in the water level and flow velocity of the Namhan River. Under these conditions, the main factors that affect the number of cyanobacteria cells are water temperature (air temperature) and light (duration of sunshine), as evidenced by the number of cyanobacteria cells showing a tendency to increase during daytime and decreasing after sunset. In addition, the number of cyanobacteria cells tend to increase at a water temperature of 20 °C or more, such as in the summer.

Cases 1–3, in which the Chungju Dam was operated using the oscillated discharge method, showed different results from the uniform discharge (Obs) (Table 5). There was a difference of 10–15 cm in the water level between Obs case and Cases 1–3, indicating that no significant difference in water level occurred as a result of a change in the discharge method. In contrast, the average flow velocity under the oscillation flow was 5–10 cm/s lower on average compared to that in the Obs case, with the standard deviation being more than 50% of the average flow velocity, indicating significant changes in flow velocity (Table 5).

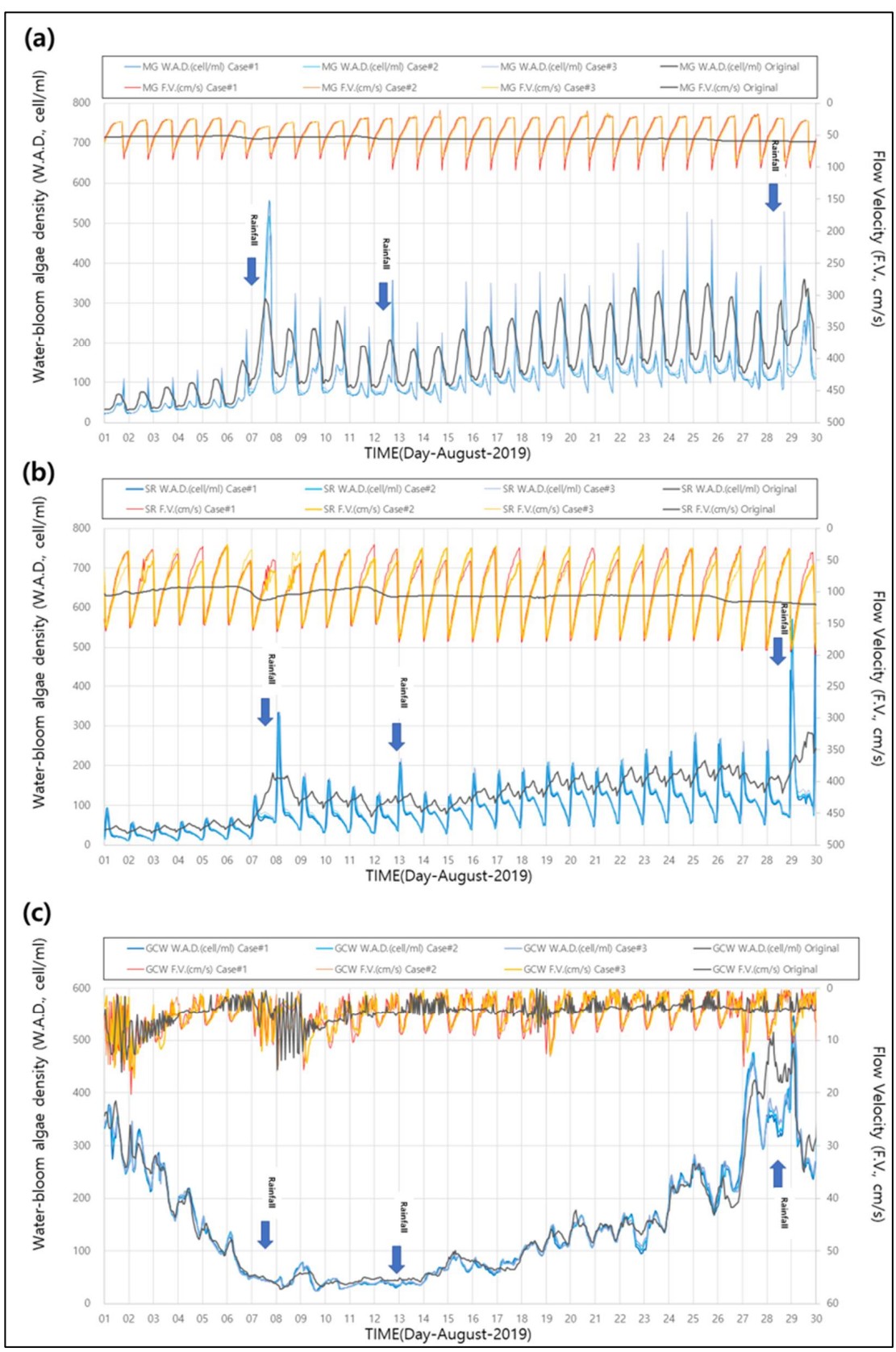

**Figure 6.** Simulation result of oscillation flow with EFDC-NIER model: (**a**) Mokgye, (**b**) Seom river junction, (**c**) Gangcheon Weir.

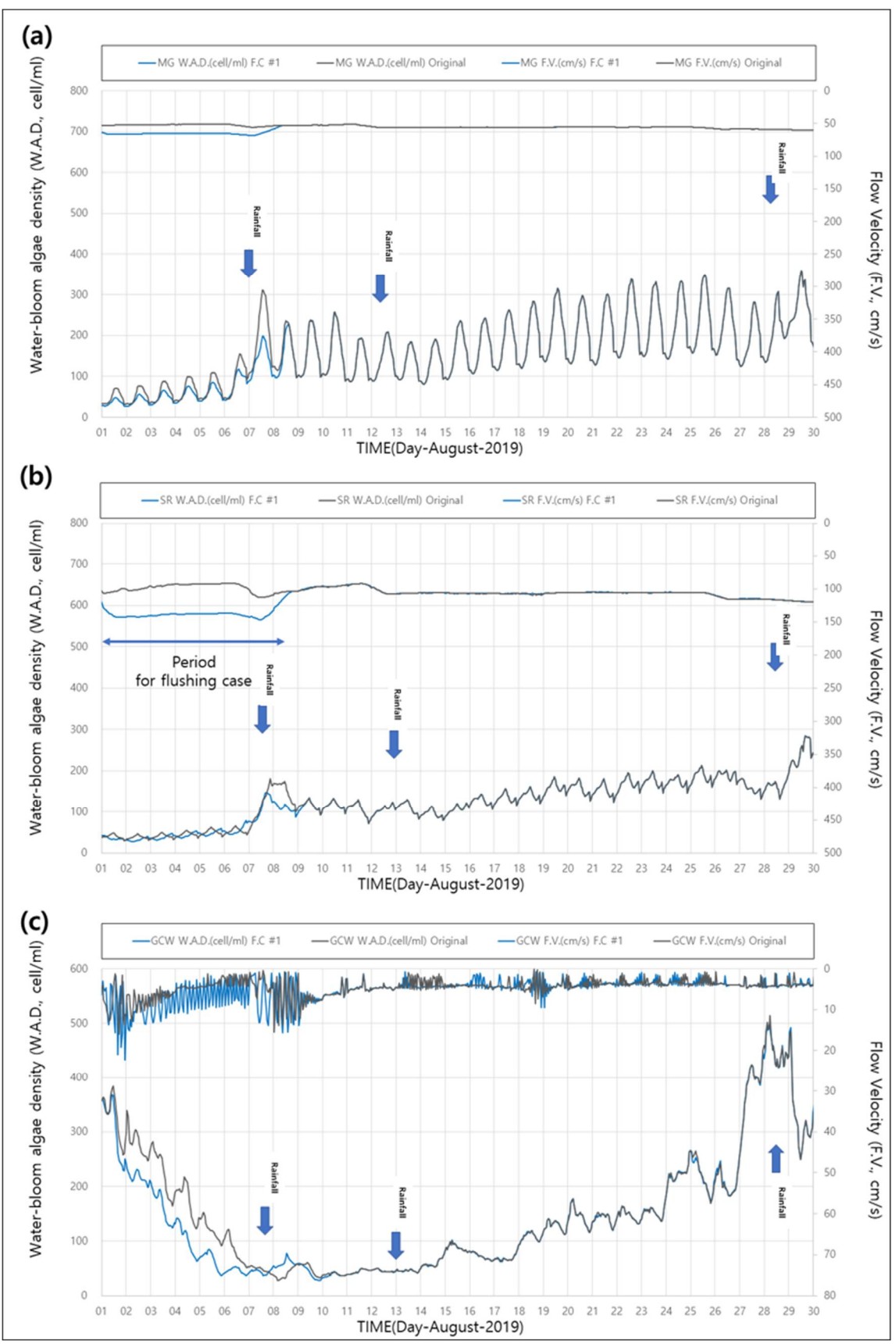

**Figure 7.** Simulation results of flushing flow with EFDC-NIER model: (**a**) Mokgye, (**b**) Seom river junction, (**c**) Gangcheon Weir.

**Table 5.** Simulation result of EFDC-NIER.

| Item | | | Discharge Rate (m³/s) | | | | |
|---|---|---|---|---|---|---|---|
| | | | Obs (24 h) | Case #1 (4 h) | Case #2 (6 h) | Case #3 (8 h) | Flushing Case |
| Mokgye (Upstream) | Water level (EL.m) | Average | 52.6 | 52.5 | 52.5 | 52.5 | 52.6 |
| | | (Standard deviation) | 0.04 | 0.33 | 0.31 | 0.28 | 0.08 |
| | Flow velocity (cm/s) | Average | 55.5 | 44.8 | 46.1 | 47.4 | 58.5 |
| | | (Standard deviation) | 2.6 | 21.9 | 21.4 | 20.1 | 4.5 |
| | Number of cyanobacteria cells (cell/mL) | Average | 163.7 | 112.8 | 117.5 | 121.6 | 159.0 |
| | | (Standard deviation) | 78.7 | 84.6 | 91.6 | 93.5 | 80.4 |
| | | Variation (%) | - | ▼31.1 | ▼28.3 | ▼25.7 | ▼2.9 |
| | | (Under discharge period) | | | | | (▼20.1) |
| Seom river junction (Midstream) | Water level (EL.m) | Average | 39.9 | 39.9 | 39.9 | 39.9 | 39.9 |
| | | (Standard deviation) | 0.04 | 0.13 | 0.13 | 0.12 | 0.06 |
| | Flow velocity (cm/s) | Average | 105.7 | 92.7 | 95.2 | 95.2 | 115.0 |
| | | (Standard deviation) | 7.7 | 43.3 | 41.7 | 40.4 | 14.6 |
| | Number of cyanobacteria cells (cell/mL) | Average | 127.4 | 90.3 | 90.5 | 92.9 | 125.3 |
| | | (Standard deviation) | 53.4 | 66.9 | 62.8 | 61.3 | 53.8 |
| | | Variation (%) | - | ▼29.1 | ▼28.9 | ▼27.1 | ▼1.6 |
| | | (Under discharge period) | | | | | (▼9.9) |
| Gangcheon weir (Downstream) | Water level (EL.m) | Average | 38.0 | 38.1 | 38.1 | 38.1 | 38.4 |
| | | (Standard deviation) | 0.06 | 0.08 | 0.08 | 0.07 | 0.08 |
| | Flow velocity (cm/s) | Average | 4.5 | 4.7 | 4.7 | 4.6 | 5.1 |
| | | (Standard deviation) | 2.2 | 3.2 | 3.1 | 3.0 | 2.9 |
| | Number of cyanobacteria cells (cell/mL) | Average | 164.4 | 162.8 | 162.2 | 162.5 | 154.7 |
| | | (Standard deviation) | 118.4 | 119.5 | 117.8 | 117.0 | 117.3 |
| | | Variation (%) | - | ▼1.0 | ▼1.4 | ▼1.2 | ▼5.9 |
| | | (Under discharge period) | | | | | (▼24.8) |

Therefore, there was no significant difference in water level between conventional uniform discharge (Obs case) and oscillation flow (Cases 1–3) but there was a significant difference in flow velocity, mainly before and after discharge.

The flow velocity significantly changed from Mokgye to the Seom River Junction in accordance with the discharge of the Chungju Dam. At Mokgye and the Seom River Junction, constant flow velocities of 55 and 105 cm/s were observed in Obs case, respectively. Under oscillation flow, the flow velocity varied from 25 to 65 cm/s at Mokgye and from 50 to 140 cm/s at the Seom River Junction, depending on the time of oscillated discharge. The number of cyanobacteria cells also differed in each section. The average monthly number of cyanobacteria cells in each case decreased by 25–31% at Mokgye and by 27–29% at the Seom River Junction compared to Obs case. In the Gangcheon Weir section, downstream of the target section, flow velocity, water level, and cyanobacteria exhibited no significant differences between Obs and the other cases. The Gangcheon Weir section was not affected by oscillation flow because the flow was adjusted by the Gangcheon Weir structure. Although the effect could be maintained in this section by operating the Gangcheon Weir in the same way as the Chungju Dam, the operation of the multifunctional weir was excluded in this study because its continuous or repeated operation may fatigue the structure or lead to accidents. Moreover, according to the water quality simulation, the maximum number of cyanobacteria cells at the hourly scale under oscillation flow become occasionally larger than in the Obs case (Figure 6). This is likely because algae rapidly and temporarily grew at the end of oscillation flow owing to a rapid reduction in the flow velocity of the river. The number of cyanobacteria cells decreased again at night as they flowed along the river, and oscillation flow began again at the time when algae rapidly increased during daytime, resulting in a reduction in the average number of cyanobacteria cells. Therefore, before the Gangcheon Weir section, the average number of cyanobacterial

cells was reduced by approximately 25–30%, and the rapid change in flow velocity caused by oscillation flow was determined to be the main cause.

As a control group for oscillation flow, flushing flow (Flushing case) that used approximately 30.2 million m$^3$ of water over seven days was simulated, and the results are shown in Figure 6. Considering the trend of the number of cyanobacteria cells in August 2019, a flushing case in which 50 m$^3$/s was additionally discharged for seven days from August 1$^{st}$ was simulated. The simulation results showed that the additional flushing flow increased the average flow velocity by approximately 10 cm/s from 55 to 65 cm/s at Mokgye. The flow velocity was maintained during the discharge period (seven days) and returned to a velocity similar to that of the Obs case after the completion of the discharge. During the flushing flow period, the number of cyanobacteria cells decreased by 20% on average at Mokgye, 10% in the Seom River, and 25% in the Gangcheon Weir section. However, the number of cyanobacteria cells showed a tendency to slightly increase after the flushing flow period, resulting in an overall reduction of 2–6% during the entire period. These results are consistent with the results of previous studies on flushing flow [25,28], indicating that the number of cyanobacteria cells decreases during the discharge period because of an increase in flow rate and velocity, returning to the original state upon the completion of discharge. Since flushing flow continuously decreases the number of cyanobacteria cells without the temporary growth of cyanobacteria (unlike oscillation flow), it is applicable for rapidly reducing algae.

## 4. Discussion

In this study, we hypothesized that algae will be reduced if turbulent or similar irregular flow occurs in a river according to the periodic change in flow rate and velocity, and the applicability of oscillation flow was evaluated. To this end, the Han River system was modeled using EFDC-NIER, which is used by ME for water quality forecasting. The oscillation flow (Cases 1–3) was simulated for summer 2019, and the results were compared with those of uniform discharge (Obs case) and flushing flow (Flushing case). Since all conditions in the simulation were constant except for the discharge from the Chungju Dam, changes in the number of cyanobacteria cells were due to the effects of dam discharge. At Mokgye and the Seom River Junction, the monthly average number of cyanobacteria cells decreased by 25–30%. This was likely due to the inhibiting effect of the irregular flow on the growth of algal colonies, which was caused by the periodic change in flow rate and velocity [37]. Therefore, the hypothesis of this study can be considered persuasive, and the proposed method is expected to reduce the number of cyanobacteria cells by 25–30% without the use of additional water resources from dam storage for treating algal blooms. Considering that a recently performed flushing flow used 20–50 million m$^3$ of water, the oscillation flow method would have a long-term mitigation effect while conserving important water resources.

Despite the use of additional water resources from dam storage, flushing flow only reduced the number of cyanobacteria cells by less than 6% per month on average (Figure 7), thereby verifying the relative effectiveness of oscillation flow (20–30%). Oscillation flow reduces the total amount of algae as it reduces the average number of cells through temporary increases in the number of cyanobacteria cells. Therefore, the proposed oscillation flow is suitable for reducing algae through constant implementation when green algae is gradually increasing in summer season. In contrast, flushing flow constantly reduces the number of cyanobacteria cells owing to the discharge of dam storage, and is thus an effective method for rapidly reducing cyanobacteria in response to forecasts or outbreaks of algae blooms. In the future, oscillation flow is expected to be applicable for diverse responses to algae if its detailed mechanism is identified through additional research.

In addition, the hypothesis of this study can be examined through Figure 8. The number of cyanobacteria cells was reduced by 20–30% depending in Cases 1–3 until the Seom River Junction, but the effect rapidly decreased downstream of the Gangcheon Weir, which exhibited a similar water level, flow velocity, number of cyanobacteria cells, and

reduction rate to those under Obs case with uniform discharge. This is because the oscillation flow transmitted from the upstream was adjusted by the Gangcheon Weir structure, making the flow downstream of this weir similar to that under uniform discharge (Obs). As shown in Figure 8, the flow velocity varied from 25 to 65 cm/s at Mokgye compared to 55 cm/s under uniform discharge (Obs) and from 50 to 140 cm/s at the Seom River Junction compared to 105 cm/s under uniform discharge (Obs). In the Gangcheon Weir section, there was no significant difference in flow velocity as it ranged from 2.3 to 6.7 cm/s under the uniform discharge (Obs) and from 1.5 to 7.9 cm/s under oscillation flow. The reduction in cyanobacteria cells was diminished in the location where the flow velocity did not significantly differ between the oscillated (Cases 1–3) and uniform discharge (Obs) cases; this is important for evaluating the hypothesis of this study. In conclusion, the concentration of cyanobacteria changes according to the fluctuation of the flow velocity for the same quantity of water.

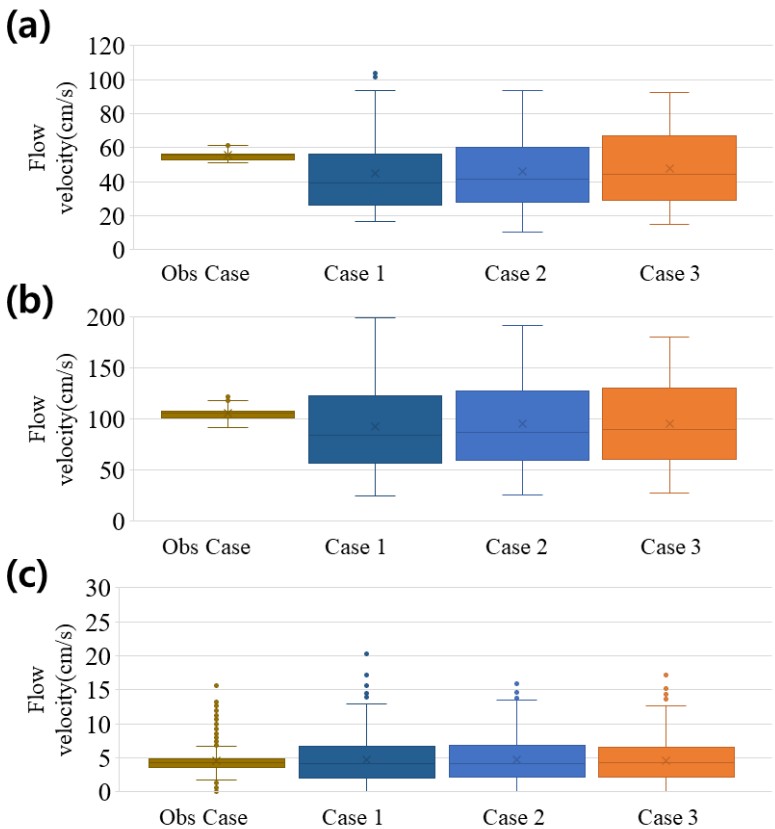

**Figure 8.** Flow velocity simulation results of the EFDC-NIER model: (**a**) Mokgye, (**b**) Seom River junction, (**c**) Gangcheon weir.

This study is also significant in that it presents a new perspective on the growth of cyanobacteria. Previous studies on responses to algal blooms in rivers mostly focused on the blocking of inflow of nutrients from point or nonpoint pollution sources, removing algae through physical or chemical means, or the temporary removal of algal blooms using flushing flow from dam storage for serious outbreaks of cyanobacteria. In this study, however, a novel method of responding to algal blooms through the flow state that could occur through the operation of artificial hydraulic structures, such as dams and weirs, was presented. However, follow-up studies are required before the practical application of the proposed concept. A clearer analysis is required of the causal relationships outlined by the hypothesis and results of this study. For example, in the Gangcheon Weir section, the reduction in the cyanobacteria cells was insignificant even though small periodic changes in flow rate and flow velocity were observed, as shown in Figure 8c. Therefore, practical

reductions only occur when the periodic changes in flow rate and velocity exceed a certain level in magnitude. To determine the level, however, large-scale analysis and verification are required. Consequently, for additional clarity about how the inhibition mechanism of the suggested concept and method, a pilot study at an existing dam in algae bloom season should be conducted. Moreover, further study of implementation of the method in practice should be performed considering hydrological conditions, water quality, and operation rules of the dam. This study aims to encourage further research by presenting information that could lead to dam operation that could respond to algal blooms without the burden of using additional dam storage.

Another limitation of this study is that algal blooms in the Han River are not severe. In the Nakdong River, the number of cyanobacteria cells tends to exceed 10,000 cell/mL every summer, thereby attracting attention as a serious social problem. In the case of the Han River, however, the number of cyanobacteria cells typically reaches only 100–200 cell/mL, and thus, the negative effects of algal blooms are relatively low compared to other major rivers. Nevertheless, the Han River was targeted in this study considering recent concerns over the increasing algal blooms in the Namhan River. Therefore, follow-up research involving the collection of additional data and verification is required. Considering these limitations, it is difficult to discuss the applicability of the proposed method using only the results of this study. However, this study is significant in that it simulated the result of a novel approach for responding to algal blooms solely through the operation of hydraulic structures. It is expected that the proposed method will become a practical countermeasure through follow-up research.

## 5. Conclusions

This study presents information to propose a dam operation that can respond to algal blooms using oscillation flow due to the periodic change in flow rate and flow velocity without the burden of using additional dam storage in summer and to encourage further research. In addition, changes in the number of cyanobacteria cells resulting from the dam operation were simulated for the Namhan River from the downstream area of the Chungju Dam to the Gangcheon Weir. The EFDC-NIER model, which is used for water quality forecasting, was constructed for the study area. Algae counts were simulated for August 2019, the month when the largest number of cyanobacteria cells have been reported in recent years, and the results were classified into five cases for simulation. The research results showed that the method proposed in this study reduced the average number of cyanobacteria cells in the Namhan River by 20–30%. The main results of this study can be summarized as follows:

1.  The reduction in algae caused by periodic changes in flow rate and velocity was hypothesized considering that turbulent flow severely inhibits the growth of phytoplankton and that algae has a tendency to reduce because of increases in river flow. Therefore, the concept of oscillation flow was proposed.

2.  For the Han River, the EFDC-NIER model, which is used for water quality forecasting, was constructed and calibrated. The constructed model was used to simulate the number of cyanobacteria cells in downstream areas of the Chungju Dam using operating oscillation flow for August 2019. The simulation results showed that there was no significant difference in water level between conventional uniform discharge (Obs) and oscillation flow but that there were differences in flow velocity, mainly before and after discharge. At Mokgye and the Seom River Junction, constant flow velocities of 55 and 105 cm/s, respectively, were observed under uniform discharge (Obs). Under oscillation flow, however, the flow velocity significantly varied from 25–65 cm/s at Mokgye and 50–140 cm/s at the Seom River Junction. Accordingly, the monthly average number of cyanobacteria cells decreased by 25–30% at Mokgye and 27–29% at the Seom River Junction. In the Gangcheon Weir section (downstream of the target section), flow velocity, water level, and number of cyanobacteria exhibited

no significant difference in the simulations by dam operation. These results indicate that changes in flow velocity are important for reducing algal growth.

3. Flushing flow was also simulated under the same conditions; the number of cyanobacteria cells decreased by 10–20% during the discharge period but reverted to the original state upon the completion of discharge, with a monthly average reduction ranging from 2 to 6%.

The most important point of the study is that it proposes a new hypothesis, concept, and method for responding to algal blooms. Since the contents of this study need to be supplemented and verified before practical application, follow-up studies on the proposed concept by hydraulic and environmental engineers are encouraged.

**Supplementary Materials:** The following supporting information can be downloaded at: https://www.mdpi.com/article/10.3390/w14081315/s1.

**Author Contributions:** Conceptualization, J.K. (Jungwook Kim) and J.K. (Jaewon Kwak); Data curation, J.K. (Jungwook Kim), J.K. (Jaewon Kwak), J.M.A., H.K., and J.J.; Formal analysis, J.K. (Jungwook Kim); Funding acquisition, K.K.; Investigation, J.K. (Jungwook Kim), J.K. (Jaewon Kwak), J.M.A., H.K., and J.J.; Methodology, J.K. (Jungwook Kim) and J.K. (Jaewon Kwak); Project administration, J.K. (Jungwook Kim) and J.K. (Jaewon Kwak); Resources, J.M.A., H.K., and K.K.; Software, J.K. (Jungwook Kim) and J.J.; Supervision, J.K. (Jaewon Kwak); Validation, J.J.; Visualization, J.M.A.; Writing—original draft, J.K. (Jungwook Kim); Writing—review and editing, J.K. (Jaewon Kwak), H.K., and K.K. All authors have read and agreed to the published version of the manuscript.

**Funding:** This research was funded by the National Institute of Environmental Research (NIER), which is funded by the Ministry of Environment (MOE) of the Republic of Korea, grant number NIER-2020-01-01-012.

**Institutional Review Board Statement:** Not applicable.

**Informed Consent Statement:** Not applicable.

**Data Availability Statement:** The data presented in this study are available on request from the corresponding author.

**Acknowledgments:** This research was funded by the National Institute of Environmental Research of South Korea.

**Conflicts of Interest:** The authors declare no conflict of interest.

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
