# Peer review of "Oscillation Flow Dam Operation Method for Algal Bloom Mitigation"

_water, doi:10.3390/w14081315_

Round 1

Reviewer 1 Report

The proposed method is based on physical force of oscillation flow, which is interesting. However, I have a concern. The algae are not dead and removed from water sources. It seems they are “diluted and temporary disappeared.” It did not solve the problem but delayed the problem. How does the author think about it? The innovation should be highlighted. The resolution of the figure should be enhanced, especially the number in the x/y axis. I suggest a minor revision to this manuscript.

Reviewer 2 Report

In the present paper, the authors used the Environmental Fluid Dynamics Code-National Institute of Environment Research (EFDC-NIER) model was constructed and calibrated for the Namhan River, South Korea. The focus of the paper is to demonstrate how the oscillation flow from the dam can help in better decreasing the concertation of algal (i.e., the algal bloom density) in the river downstream of the dam. It was found that the proposed oscillation flow operation leads to a decrease of about 30% of the algal bloom density. This is a very interesting investigation, new and original, and few similar studies are available in the literature.  The paper is well written, and scientifically sound. Few amendments are necessary before the final acceptation. 

  1. Literature review is incomplete and needs improvement, especially the state or the art and why the proposed method has been introduced.
  2. Statistical description of the water quality variables used in the present study is necessary.
  3. Boundary condition of EFDC-NIER model in Han river are not well justified and how they are selected.
  4. Results of the model calibration are presented in Table 3, however, the simulation results are not presented.
  5. I can see that calibration results of the Chl-a and BOD are very poor (Table 3) which should be justified and deeply discussed.
  6. Figure 6 and 7 are for poor quality.

Round 2

Reviewer 2 Report

The authors have provided the necessary revision and the paper is ready for publication. Now further revision is necessary.